# Common protocol for validation of the QCOVID algorithm across the four UK nations

Steven Kerr [1], Chris Robertson,[2] Vahe Nafilyan [3], Ronan A Lyons [4],
Frank Kee,[5] Christopher R Cardwell,[6] Carol Coupland,[7] Jane Lyons [8],
Ben Humberstone,[9] Julia Hippisley-Cox [10] Aziz Sheikh[1]

## ABSTRACT

**Introduction** The QCOVID algorithm is a risk prediction tool for infection and subsequent hospitalisation/death due to SARS-CoV-2. At the time of writing, it is being used in important policy-making decisions by the UK and devolved governments for combatting the COVID-19 pandemic, including deliberations on shielding and vaccine prioritisation. There are four statistical validations exercises currently planned for the QCOVID algorithm, using data pertaining to England, Northern Ireland, Scotland and Wales, respectively. This paper presents a common procedure for conducting and reporting on validation exercises for the QCOVID algorithm.

**Methods and analysis** We will use open, retrospective cohort studies to assess the performance of the QCOVID risk prediction tool in each of the four UK nations. Linked datasets comprising of primary and secondary care records, virological testing data and death registrations will be assembled in trusted research environments in England, Scotland, Northern Ireland and Wales. We will seek to have population level coverage as far as possible within each nation. The following performance metrics will be calculated by strata: Harrell's C, Brier Score, $R^2$ and Royston's D.

**Ethics and dissemination** Approvals have been obtained from relevant ethics bodies in each UK nation. Findings will be made available to national policy-makers, presented at conferences and published in peer-reviewed journal.

## INTRODUCTION

The QCOVID algorithm[1] has been developed to help identify adults at high risk of being hospitalised or dying following infection with SARS-CoV-2. The algorithm takes as input a total of 40 variables including age, sex, ethnicity, Townsend Deprivation Score (TDS)[2] and housing category, as well as clinical information including body mass index (BMI) and 33 variables related to medical conditions and treatments. It outputs the predicted probability that an individual will be infected with SARS-CoV-2 and then hospitalised, and the predicted probability that an individual will be infected with SARS-CoV-2 and then die, over a 90-day period.

For numbered affiliations see end of article.

**Correspondence to**
Dr Steven Kerr;
steven.kerr@ed.ac.uk

---

### STRENGTHS AND LIMITATIONS OF THIS STUDY

⇒ We will use national level data within each UK nation.
⇒ There are potential issues with missing data and differences in the way data are recorded in each country.
⇒ We will evaluate the performance of the algorithm according to several relevant metrics.

---

The algorithm was trained using information from the QResearch database,[3] which as of April 2020 contained routinely collected data from 1205 general practices across England, covering 10.5 million patients. The initial training dataset comprised of a cohort of 6.08 million individuals tracked from 24 January 2020 to 30 April 2020, and was validated on a subset of 2.17 million individuals tracked from 1 May 2020 to 30 June 2020. The research protocol for the development of the QCOVID algorithm can be found in Hippisley-Cox *et al.*[4]

The QCOVID algorithm was commissioned by the Chief Medical Officer for England on behalf of the UK government. The algorithm has been used to inform UK and devolved government policy on combatting the SARS-CoV-2 pandemic, including guidance on social distancing and shielding measures, as well vaccine prioritisation.[5] It is therefore of great importance to validate the predictions of the algorithm in subpopulations of the UK that were not in the initial training set, but will potentially be subject to those policies.

At the time of writing, there are validation exercises planned in Scotland, Northern Ireland and Wales, and a validation exercise underway in England. Validation work was considered urgent and has been expedited in order to support national decision-making. In order to facilitate useful comparison of the results of the separate validation exercises, it is necessary to establish a consistent set of

**England:**

Office for National Statistics (ONS) Public Health Linked Data Asset. This dataset is based on the 2011 census in England covering 40.1 million people, linked at individual level using the National Health Service (NHS) number to mortality records, Hospital Episode Statistics and the General Practice Extraction Service data for pandemic planning and research. The data covers 80% of the population of England aged 19 and over.

**Northern Ireland:**

National Health Application and Infrastructure Services will be used for demographic information. The Patient Administration System will be used for data on hospital admissions. Death data will be drawn from the registrar general, and identified as COVID-19 related through the official Northern Ireland Statistics and Research Agency dashboard. The General Practice Information Platform will bring together general practice (GP) records from practices across Northern Ireland into a single dataset for use in the validation. As this is not held in the Honest Broker Service, a separate request to its governance board is being made. The Electronic Prescribing Database will be used to access information on prescriptions.

**Scotland:**

EAVE II (Early Pandemic Evaluation and Enhanced Surveillance of COVID-19) dataset.[9] Contains primary healthcare records for 5.4 million people covering 99% of the population of Scotland, linked with secondary care data from Scottish Morbidity Record, COVID-19 test results from Electronic Communication of Surveillance Scotland and mortality data from National Records Scotland.

**Wales:**

Secure Anonymised Information Linkage System.[8] This will use the Controlling COVID-19 platform linking records on 3.2 million people from the NHS population spine with hospital (Patient Episode Database for Wales), Welsh Longitudinal GP record, COVID-19 test results from the Laboratory Information Management System and mortality and 2011 census data from the ONS.[10]

procedures. The purpose of this paper is to explicate a common methodology for the validation of the QCOVID algorithm across the four nations of the UK.

## METHODS AND ANALYSIS
### Study design

Open, retrospective cohort study designs will be employed, making use of routinely collected data from general practices for clinical and demographic information, as well as linked datasets on hospital admissions, reverse transcription PCR testing for COVID-19 and registered deaths. We will aim to have national coverage as far as is possible within each of the four nations of the UK.

### Data sources

Box 1 contains a brief summary of the main datasets that will be used in the validation exercise for each nation.

### Selection criteria

Any individual in the relevant linked dataset between the ages of 19 and 100 will be included. Individuals who had

an event (hospitalisation or death) in the first period (24 January 2020–30 April 2020) will be excluded from any analysis in the second period (1 May 2020–30 June 2020).

These time periods were chosen to mirror the time periods in the original QCOVID paper. After the vaccination programme started in the UK on 8 December 2020, work had already begun on QCOVID 2 and 3, which will take into account vaccination status. Future validation work will focus on QCOVID 2 and 3 for more recent time periods.

### Exposure and outcomes

Tables 1 and 2 list all exposure and outcomes variables respectively for the QCOVID algorithm, along with a description, variable type (eg, integer, real, categorical) and possible values.

Whenever available, all variables will be taken as the most recent recorded value in the relevant dataset at the date of entry into the cohort. The TDS will be determined by matching available residential location information with output area and the corresponding TDS from the 2011 UK census.[6] Categories for the variable chemocat will be determined using the lookup table in the online supplemental materials.

### Data cleaning

The following procedures will be used for data cleaning:

► diabetes_cat: If the most recent entry has both type 1 and type 2 recorded, diabetes_cat will be set to type 2.
► BMI: The most recently recorded patient BMI within the last 5 years. If the most recently recorded BMI is from more than 5 years ago at the search date, BMI will be set to missing value. Implausible values for BMI (<12 or >70) will be set to missing value.
► learncat: If a patient is recorded has having both learning disability and Down's syndrome, learncat will be set to Down's syndrome.

### Missing data

For comorbidities and medication use and treatments, missing values will be taken to mean absence of that factor. Modal substitution will be considered for missing values for ethnicity. For any other missing values of predictor variables, a single imputation will be considered. Outcome variables will not be imputed, and nor will they be included as predictors in the imputation. The following methods may be considered for use in the imputation: predictive mean matching, least squares, logistic and multinomial models, imputation by chained equations.

### Statistical analysis

Each validation exercise will report a table of cohort characteristics, following table 2 in Clift *et al*.[1] The main performance metrics that will be calculated are $R^2$,[6] Harrell's C, Royston's D[7] and the Brier Score. Different stratifications for these statistics will be considered, including by age, sex and time period. 95% CIs will be reported for $R^2$, Harrell's C and Royston's D. Graphs of observed and

**Table 1** Exposure variables in QCOVID algorithm

| Variable | Description/question | Value |
|---|---|---|
| Demographic | | |
| age | Age in years | Integer: 19–100 |
| sex | Biological sex at birth | Categorical: female, male |
| town | Townsend Deprivation Score | Real number |
| ethnicity | Ethnicity | Categorical white, Indian, Pakistani, Bangladeshi, other Asian, Caribbean, black African, Chinese, other ethnic group |
| homecat | What is your housing category—care home or homeless or neither? | Categorical neither, care home, homeless |
| Clinical: | | |
| bmi | Body Mass Index (kg/m$^2$) | Positive real number |
| chemocat | Have you had chemotherapy in the last 12 months? | Categorical: none, group A, group B, group C |
| learncat | Do you have a learning disability or Down's syndrome? | Categorical: learning disability, Down's syndrome |
| renalcat | Chronic kidney disease (CKD) stage | Categorical: No serious kidney disease, CKD stage 3, CKD stage 4, CKD stage 5 without dialysis or transplant, CKD stage 5 with dialysis in last 12 months, CKD stage 5 with transplant |
| diabetescat | Do you have diabetes? | Categorical: none, type 1, type 2 |
| b2_82 | Have you been prescribed immunosuppressants four or more times in the previous 6 months? | Categorical: yes, no |
| b2_leukolaba | Have you been prescribed antileukotriene or long acting beta2-agonists (LABA) four or more times in the previous 6 months? | Categorical: yes, no |
| b2_prednisone | Have you been prescribed oral predisolone containing preparations prescribed four or more times in the previous 6 months? | Categorical: yes, no |
| b_AF | Do you have atrial fibrillation? | Categorical: yes, no |
| b_CCF | Do you have heart failure? | Categorical: yes, no |
| b_asthma | Do you have asthma? | Categorical: yes, no |
| b_bloodcancer | Have you a cancer of the blood or bone marrow such as leukaemia, myelodysplastic syndromes, lymphoma or myeloma and are at any stage of treatment? | Categorical: yes, no |
| b_cerebralpalsay | Do you have cerebral palsy? | Categorical: yes, no |
| b_chd | Do you have coronary heart disease? | Categorical: yes, no |
| b_cirrhosis | Do you have cirrhosis of the liver? | Categorical: yes, no |
| b_congenheart | Do you have congenital heart disease or have you had surgery for it in the past? | Categorical: yes, no |
| b_copd | Do you have chronic obstructive pulmonary disease (COPD)? | Categorical: yes, no |
| b_dementia | Do you have dementia? | Categorical: yes, no |
| b_epilepsy | Do you have epilepsy? | Categorical: yes, no |
| b_fracture4 | Have you had a prior fracture of hip, wrist, spine or humerus? | Categorical: yes, no |

**Table 1** Continued

| Variable | Description/question | Value |
|---|---|---|
| b_neurorare | Do you have motor neuron disease, multiple sclerosis, myasthenia or Huntington's chorea? | Categorical: yes, no |
| b_parkinsons | Do you have Parkinson's disease? | Categorical: yes, no |
| b_pulmhyper | Do you have pulmonary hypertension or pulmonary fibrosis? | Categorical: yes, no |
| b_pulmrare | Do you have cystic fibrosis or bronchiectasis or alveolitis? | Categorical: yes, no |
| b_pvd | Do you have peripheral vascular disease? | Categorical: yes, no |
| b_ra_sle | Do you have rheumatoid arthritis or systemic lupus erythematosus? | Categorical: yes, no |
| b_respcancer | Do you have lung or oral cancer? | Categorical: yes, no |
| b_semi | Do you have severe mental illness? | Categorical: yes, no |
| b_sicklecelldisease | Do you have sickle cell disease or severe combined immune deficiency syndromes? | Categorical: yes, no |
| b_stroke | Have you had a stroke or transient ischaemic attack? | Categorical: yes, no |
| b_vte | Have you had a thrombosis or pulmonary embolus? | Categorical: yes, no |
| p_marrow6 | Have you had a bone marrow or stem cell transplant in the last 6 months? | Categorical: yes, no |
| p_radio6 | Have you had radiotherapy in the last 6 months? | Categorical: yes, no |
| p_solidtransplant | Have you had a solid organ transplant (lung, liver, stomach, pancreas, spleen, heart or thymus)? | Categorical: yes, no |

predicted probability of hospital admission and death by vigintile for stratified subgroups will be reported, following Clift *et al.*[1]

### Sample size

A preliminary sample size calculation can be done using figures from the original paper.[1] Using the estimated SD of Harrell's C for females in the first time period and assuming Harrell's C is asymptotically normally distributed implies that a sample size of approximately 5714 would be sufficient to correctly reject a null hypothesis of C=0.5 at significance level 0.05 with probability 80% given a true value of C=0.8. Repeating this calculation for other population subgroups and time periods yields results of a similar magnitude. The samples sizes in the planned studies will be on the order of hundreds of thousands or millions.

### Ethics, reporting and dissemination

The ethics approval for the development and validation of QCOVID in England was granted by the East Midlands-Derby Research Ethics Committee (reference 18/EM/0400). For Scotland, approvals have been obtained by the National Research Ethics Service Committee (REC), South East Scotland 02 (REC number: 12/SS/0201) and the Public Benefit and Privacy Panel for Health and Social Care (reference number: 1920-0279). The data to be used in this study for Wales are available in the Secure Anonymised Information Linkage (SAIL) Databank at Swansea University, Swansea, UK. All proposals to use SAIL data are subject to review by an independent Information Governance Review Panel (IGRP). Before any data can be accessed, approval must be given by the IGRP. The IGRP gives careful consideration to each project to ensure proper and appropriate use of SAIL data. When access has been approved, it is gained through a privacy-protecting safe haven and remote access system referred to as the SAIL Gateway. SAIL has established an application process to be followed by anyone who would like to access data via SAIL.[8] Findings will be presented at conferences, published in peer-reviewed journals and to the funders and government COVID-19 advisory bodies as appropriate. Strengthening the Reporting of Observational Studies in Epidemiology and Reporting of studies Conducted using Observational Routinely collected Data

**Table 2** Outcomes variables in QCOVID algorithm

| Variable | Description/Question | Value |
|---|---|---|
| Time to Covid-19 hospitalisation | Time to hospitalisation with reverse transcription (RT) PCR confirmed Covid-19 infection in the cohort period in days. | Real number: 0–91 |
| Time to Covid-19 death | Time to death with Covid-19 confirmed or suspected on their death certificate, or confirmed by RT-PCR test, in the cohort period in days. | Real number: 0–91 |

(via the COVID-19 extension) checklists will guide our study findings reporting. The Northern Ireland validation study proposal is under review by the NITRE (Norther Ireland Trusted Research Environment) for HSC (Health and Social Care) data accessed via Northern Ireland Honest Broker Service; an ethics application has been submitted through IRAS (Integrated Research Application System).

**Author affiliations**
[1]Usher Institute, University of Edinburgh, Edinburgh, UK
[2]Department of Mathematics and Statistics, University of Strathclyde, Glasgow, UK
[3]Office for National Statistics, Newport, UK
[4]Swansea Clinical School, University of Wales Swansea, Swansea, UK
[5]UKCRC Centre of Excellence for Public Health (NI), Queen's University Belfast, Belfast, UK
[6]School of Medicine, Dentistry and Biomedical Sciences, Queen's University Belfast, Belfast, UK
[7]Division of Primary Care, University of Nottingham, Nottingham, UK
[8]Population Data Science, Swansea University Medical School, Swansea, UK
[9]Office for National Statistics, London, UK
[10]Nuffield Department of Primary Care Sciences, University of Oxford, Oxford, UK

**Acknowledgements** This work will use data provided by patients and collected by a number of organisations. We would like to acknowledge all patients who shared their information as well as all data providers who make anonymised data available for research. In particular, Public Health Scotland, Public Health Wales, Public Health England, the National Health Service, the Secure Anonymised Information Linkage databank and the Office for National Statistics.

**Contributors** AS conceived this protocol. CR, VH, FK, TC, JH-C, BH, CC, RL and JL provided country specific information about available data and analysis plans. SK wrote drafts of this protocol. All authors gave final approval of the version to be published.

**Funding** The validation in England will be funded by a grant from the National Institute for Health Research following a commission by the Chief Medical Officer for England. In Scotland, EAVE II is funded by the Medical Research Council (MR/R008345/1) and supported by the Scottish Government. In Wales, Controlling COVID-19 is supported by the Medical Research Council (MR/V028367/1).

**Competing interests** AS reports grants from NIHR, grants from MRC, and grants from HRR UK, during the conduct of the study. JL and RL report grants from UKRI Medical Research Council, during the conduct of the study. JH-C reports grants from John Fell Oxford University Press Research Fund, grants from Cancer Research UK (CR-UK) grant number C5255/A18085, through the Cancer Research UK Oxford Centre, grants from the Oxford Wellcome Institutional Strategic Support Fund (204826/Z/16/Z), grants from NIHR, during the conduct of the study; personal fees and other from ClinRisk, outside the submitted work; and JH-C is an unpaid director of QResearch, a not-for-profit organisation which is a partnership between the University of Oxford and EMIS Health who supply the QResearch database used for this work. Carol Coupland reports personal fees from ClinRisk, outside the submitted work. JH-C, AS and CC were members of the research team involved in the development of the QCOVID risk prediction algorithm. All other authors report no conflict of interest

**Patient and public involvement** Patients and/or the public were not involved in the design, or conduct, or reporting, or dissemination plans of this research.

**Patient consent for publication** Not applicable.

**Provenance and peer review** Not commissioned; externally peer reviewed.

**ORCID iDs**
Steven Kerr http://orcid.org/0000-0002-3643-7859
Vahe Nafilyan http://orcid.org/0000-0003-0160-217X
Ronan A Lyons http://orcid.org/0000-0001-5225-000X
Jane Lyons http://orcid.org/0000-0002-4407-770X
Julia Hippisley-Cox http://orcid.org/0000-0002-2479-7283

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
