## [Reviewer comments · BMJ Open]

ARTICLE DETAILS

TITLE (PROVISIONAL)	A common protocol for validation of the QCOVID algorithm across the four UK nations
AUTHORS	Kerr, Steven; Robertson, Chris; Nafilyan, Vahe; Lyons, Ronan; Kee, Frank; Cardwell, Christopher; Coupland, Carol; Lyons, Jane; Humberstone, Ben; Hippisley-Cox, Julia; Sheikh, Aziz

VERSION 1 – REVIEW

REVIEWER	McCaffrey, Peter UTMB
REVIEW RETURNED	17-May-2021

GENERAL COMMENTS	One small possible type-o I noticed on page 9 of 14, line 30. in the "Description/Question" column corresponding to the item b_neurorare.
---

REVIEWER	Sperrin, Matthew University of Manchester Institute of Population Health
REVIEW RETURNED	02-Jun-2021

GENERAL COMMENTS	This paper describes the common methodology that will be followed when validating QCOVID within the UK. Having a common protocol for multiple validations is an excellent idea and I commend the rigour. However I do have some concerns about the execution. The first is that at least two of the validations described, appear to predate this protocol by several months. The second is the detail in the analysis plan part of the protocol is, in places, too vague to really be useful. Specific comments follow. Comments: 1. There are existing validations of QCOVID that predate this protocol (from the same group of authors) - including Nafilyan et al (https://www.thelancet.com/journals/landig/article/PIIS2589-7500(21)00080-7/fulltext), and Simpson et al (https://papers.ssrn.com/sol3/papers.cfm?abstract_id=3805856). Both of these were completed some time ago (e.g. the Nafilyan paper was available since Jan 2021, while this submission is from May 2021) - so I am unclear how they relate to the present protocol, which is written in the future tense and does not refer to these papers.2. The two periods of analysis both end by June 2020. Given the fast changing situation is it unclear why (assuming that this paper really is reporting planned future validations) more recent periods, such as winter 2020/21, would not be included?3. BMI data cleaning: will this not be range-checked, e.g. exclude values <12 or >70? There have been high profile cases where
--

	implausible values of BMI have generated surprising QCOVID predictions. 4. Missing data: further information is required. If this is a protocol should it not be more definitive about the strategy that will be used rather than a list of approaches that 'may be considered'? Will outcome be included in the imputation models or not (there are good reasons for doing each e.g. see http://onlinelibrary.wiley.com/doi/abs/10.1002/bimj.201400004) 5. Statistical analysis section is vague. E.g. 'Other analyses/reporting measures will also be considered.' What other analyses and measures? This is a protocol so specificity is expected. 6. Sample size: I am not clear what is meant by the sample size calculation. Surely not a test of whether Harrell's C is different from 0.5, which would be somewhat uninformative? Please give references, e.g. https://pubmed.ncbi.nlm.nih.gov/26553135/. There is also more recently https://t.co/40Vrln57G1?amp=1 - but I appreciate that this postdates the protocol so I wouldn't expect it to be used. 7. I would expect some mention of reporting checklists that will be adhered to in the planned studies, e.g. TRIPOD, as well as some commitment to transparency of analysis code used (I appreciate, of course, that the data cannot be made available)
--	---

REVIEWER	Mao, Zijun Huazhong University of Science and Technology, College of Public Administration
REVIEW RETURNED	26-Oct-2021

GENERAL COMMENTS	Total comment The Protocol introduced the QCOVID algorithm, which has a wide range of application scenarios in predicting hospitalization and death from COVID-19. INTRODUCTION 1. It is necessary to compare the QCOVID algorithm with other similar algorithms to determine the advantages of the QCOVID algorithm. METHODS AND ANALYSIS 2. The variables shown in Table 1 need to indicate which database they come from. 3. Among clinical variables, the prevalence of some diseases is extremely low, and there is probably zero inflation, which affects the accuracy of the model. The question is, how to adjust methods and algorithms to properly handle these variables? 4. The abbreviation needs to be explained after the text.
--

VERSION 1 – AUTHOR RESPONSE

Reviewer: 1

Dr. Peter McCaffrey, UTMB

Comments to the Author:

One small possible typo I noticed on page 9 of 14, line 30. in the "Description/Question" column corresponding to the item b_neurorare.

Response: Thank you for pointing this out. We have fixed the typo by changing “myaesthesia” to “myasthenia”.

Reviewer: 2

Dr. Matthew Sperrin, University of Manchester Institute of Population Health

Comments to the Author:

This paper describes the common methodology that will be followed when validating QCOVID within the UK. Having a common protocol for multiple validations is an excellent idea and I commend the rigour. However I do have some concerns about the execution. The first is that at least two of the validations described, appear to predate this protocol by several months. The second is the detail in the analysis plan part of the protocol is, in places, too vague to really be useful. Specific comments follow.

Comments:

1. There are existing validations of QCOVID that predate this protocol (from the same group of authors) - including Nafilyan et al ([https://www.thelancet.com/journals/landig/article/PIIS2589-7500\(21\)00080-7/fulltext](https://www.thelancet.com/journals/landig/article/PIIS2589-7500(21)00080-7/fulltext)), and Simpson et al (https://papers.ssrn.com/sol3/papers.cfm?abstract_id=3805856). Both of these were completed some time ago (e.g. the Nafilyan paper was available since Jan 2021, while this submission is from May 2021) - so I am unclear how they relate to the present protocol, which is written in the future tense and does not refer to these papers.

Response: Work on this protocol began in November 2020, was substantively complete by January 2021, and was submitted on 5 March 2021. Validation work was considered urgent and was expedited. The validation by Nafilyan et al partly overlapped with work on the protocol. The validation by Simpson et al was carried out after this protocol was finished.

2. The two periods of analysis both end by June 2020. Given the fast changing situation is it unclear why (assuming that this paper really is reporting planned future validations) more recent periods, such as winter 2020/21, would not be included?

Response: These time periods were chosen to mirror the time periods in the original QCovid algorithm paper. After the vaccination programme started in the UK on 8 December 2020, work had already begun on QCovid 2&3, which take into account vaccination status. Our intention was to focus on validation of QCovid 2&3 for more recent time periods.

3. BMI data cleaning: will this not be range-checked, e.g. exclude values <12 or >70? There have been high profile cases where implausible values of BMI have generated surprising QCOVID predictions.

Response: Thank you for raising this good point. There were no implausible values for BMI in the two validation exercises that have been carried out to date – this field was cleaned before it arrived in the trusted research environments. We have added the following text to the data cleaning section for future validation work:

'Implausible values for BMI (<12 or >70) will be set to missing value.'

4. Missing data: further information is required. If this is a protocol should it not be more definitive about the strategy that will be used rather than a list of approaches that 'may be considered'? Will outcome be included in the imputation models or not (there are good reasons for doing each e.g. see <http://onlinelibrary.wiley.com/doi/abs/10.1002/bimj.201400004>)

Response: We purposely allowed for the possibility of several imputation strategies, because the extent/nature of missing data, and therefore the optimal strategy for dealing with it, may vary by country. Covid tests, hospitalisation and deaths tends to be well recorded, so we did not consider imputing outcome. We have added the following text to the 'Missing data' section:

'For any other missing values of predictor variables, a single imputation will be considered. Outcome variables will not be imputed.'

5. Statistical analysis section is vague. E.g. 'Other analyses/reporting measures will also be considered.' What other analyses and measures? This is a protocol so specificity is expected.

Response: We agree that this is excessively vague. We have removed that sentence.

6. Sample size: I am not clear what is meant by the sample size calculation. Surely not a test of whether Harrell's C is different from 0.5, which would be somewhat uninformative? Please give references, e.g. <https://pubmed.ncbi.nlm.nih.gov/26553135/>. There is also more recently <https://t.co/40Vrln57G1?amp=1> - but I appreciate that this postdates the protocol so I wouldn't expect it to be used.

Response: The sample size calculation refers to the sample size required to detect a value for Harrell's C greater than or equal to 0.8, with 80% power. We thank you for pointing us towards these references; however they were not used to inform the sample size calculation.

7. I would expect some mention of reporting checklists that will be adhered to in the planned studies, e.g. TRIPOD, as well as some commitment to transparency of analysis code used (I appreciate, of course, that the data cannot be made available)

Response: In the Ethics, reporting and dissemination section, the protocol reads

'Strengthening the Reporting of Observational Studies in Epidemiology (STROBE) and Reporting of studies Conducted using Observational Routinely-collected Data (RECORD) (via the COVID-19 extension) checklists will guide our study findings reporting.'

Publicly available GitHub repositories with the code used in validation work have been provided in both validation papers that have been published to date. We have added a data sharing statement at the end of the protocol,

'Data sharing:

All code used in these analyses will be made publicly available online e.g. through GitHub.'

Reviewer: 3

Prof. Zijun Mao, Huazhong University of Science and Technology, Huazhong University of Science and Technology

Comments to the Author:

Total comment

The Protocol introduced the QCOVID algorithm, which has a wide range of application scenarios in predicting hospitalization and death from COVID-19.

INTRODUCTION

1. It is necessary to compare the QCOVID algorithm with other similar algorithms to determine the advantages of the QCOVID algorithm.

Response: The scope of the work detailed in this protocol was validation only. There was no intention to compare QCovid with other algorithms.

METHODS AND ANALYSIS

2. The variables shown in Table 1 need to indicate which database they come from.

Response: We have amended the text in the 'Study design' section as follows:

'Open, retrospective cohort study designs will be employed, making use of routinely collected data from General Practices for clinical and demographic information, as well as linked datasets on hospital admissions, reverse-transcription polymerase chain reaction (RT-PCR) testing for Covid-19, and registered deaths. We will aim to have national coverage as far as is possible within each of the four nations of the UK.'

This, together with the information in Box 1, indicates the data sources used in each nation.

3. Among clinical variables, the prevalence of some diseases is extremely low, and there is probably zero inflation, which affects the accuracy of the model. The question is, how to adjust methods and algorithms to properly handle these variables?

Response: The work detailed in this protocol entails validation of the QCovid algorithm only. There was no intention to modify the algorithm in any way. For comorbidity and treatment variables, missing values were taken to indicate absence of the comorbidity/treatment. We believe this is reasonable because we believe that they are typically only recorded if they are present.

4. The abbreviation needs to be explained after the text.

Response: We have amended the text to spell out all acronyms the first time they appear.

We are grateful for the opportunity to revise our protocol in the light of this thoughtful and constructive feedback. Please do not hesitate to contact us if you require any further revisions or clarification.

With kind regards,

Aziz Sheikh and Steven Kerr, on behalf of the co-authors

VERSION 2 – REVIEW

REVIEWER	Sperrin, Matthew University of Manchester Institute of Population Health
REVIEW RETURNED	07-Dec-2021

GENERAL COMMENTS	Thanks to the authors for their responses to my comments. Some issues, however, are still outstanding, as detailed below. - I appreciate the clarification regarding my points 1 and 2. I feel it would be useful to add an explanatory note to the protocol, explaining both of these issues to the reader. - On point 4, to reiterate, the question is whether outcome will be included in the imputation models, not whether outcome will be imputed. The authors have not responded to that.
--

	- On point 6, further clarity is still needed. The response is "the sample size required to detect a value for Harrell's C greater than or equal to 0.8, with 80% power." What does it mean 'to detect'? The talk of power suggests that 'to detect' means a null hypothesis being rejected - what was this null hypothesis? I assume it is not that Harrell's C = 0.5?
--	---

REVIEWER	Mao, Zijun Huazhong University of Science and Technology, College of Public Administration
REVIEW RETURNED	17-Jan-2022

GENERAL COMMENTS	After reviewing this manuscript, I believe that this version has been appropriately revised and clearly answered my questions. I think it meets the standards of publication and recommends direct publication.
---

VERSION 2 – AUTHOR RESPONSE

Review 2:

- I appreciate the clarification regarding my points 1 and 2. I feel it would be useful to add an explanatory note to the protocol, explaining both of these issues to the reader.

The following text is at the end of the Introduction (Page 3, Para 3):

'At the time of writing, there are validation exercises planned in Scotland, Northern Ireland and Wales, and a validation exercise underway in England. Validation work was considered urgent and has been expedited in order to support national decision making.'

The following text has been added to the selection criteria section (Page 5, Para 1):

'These time periods were chosen to mirror the time periods in the original QCOVID paper. After the vaccination programme started in the UK on 8 December 2020, work had already begun on QCOVID 2&3, which will take into account vaccination status. Future validation work will focus on QCOVID 2&3 for more recent time periods.'

- On point 4, to reiterate, the question is whether outcome will be included in the imputation models, not whether outcome will be imputed. The authors have not responded to that.

The Missing Data section has been amended as follows:

'Outcome variables will not be imputed, and nor will they be included as predictors in the imputation.'

- On point 6, further clarity is still needed. The response is "the sample size required to detect a value for Harrell's C greater than or equal to 0.8, with 80% power." What does it mean 'to detect'? The talk of power suggests that 'to detect' means a null hypothesis being rejected - what was this null hypothesis? I assume it is not that Harrell's C = 0.5?

We have amended the Sample Size section as follows (Page 10, Para 1):

'Using the estimated standard deviation of Harrell's C for females in the first time period and assuming Harrell's C is asymptotically normally distributed implies that a sample size of approximately 5,714 would be sufficient to correctly reject a null hypothesis of C=0.5 at significance level 0.05 with probability 80% given a true value of C=0.8'

Thank you for the opportunity to revise our protocol in the light of this thoughtful and constructive feedback. Please do not hesitate to contact us if you require any further revisions or clarification.

With kind regards,

Aziz Sheikh and Steven Kerr, on behalf of the co-authors